# Clinical Usefulness of Immune Profiling for Differential Diagnosis between Crohn’s Disease, Intestinal Tuberculosis, and Behcet’s Disease

**DOI:** 10.3390/diagnostics13182904

**Published:** 2023-09-11

**Authors:** Ji Won Yoo, Su In Jo, Dong Woo Shin, Ji Won Park, Sung-Eun Kim, Hyun Lim, Ho Suk Kang, Sung-Hoon Moon, Min Kyu Kim, Sang-Yeob Kim, Sung Wook Hwang, Jae Seung Soh

**Affiliations:** 1Department of Internal Medicine, Kirk Kerkorian School of Medicine, University of Nevada, Las Vegas, NV 89557, USA; ji.yoo@unlv.edu; 2PrismCDX Co., Ltd., Hwaseong-si 18469, Republic of Korea; gussue@naver.com; 3Department of Internal Medicine, Hallym University Sacred Heart Hospital, University of Hallym College of Medicine, Anyang 14068, Republic of Korea; delight0618@hallym.or.kr (D.W.S.); miunorijw@hallym.or.kr (J.W.P.); sekim@hallym.or.kr (S.-E.K.); hlim77@hallym.or.kr (H.L.); hskang76@hallym.or.kr (H.S.K.); endomoon@hallym.or.kr (S.-H.M.); 4Department of Gastroenterology, Asan Medical Center, University of Ulsan College of Medicine, Seoul 05505, Republic of Korea; min4211@naver.com; 5Convergence Medicine Research Center, Asan Institute for Life Sciences, Asan Medical Center, Seoul 05505, Republic of Korea; sykim3yk@amc.seoul.kr

**Keywords:** immune cell, inflammatory bowel disease, immunohistochemistry, quantitative evaluation, differential diagnosis

## Abstract

It is important to make a differential diagnosis between inflammatory diseases of the bowel with similar clinical and endoscopic features. The profiling of immune cells could be helpful for accurately diagnosing inflammatory bowel diseases. We compared immune marker expression between Crohn’s disease (CD), intestinal Behcet’s disease (BD), and intestinal tuberculosis (TB) and evaluated the usefulness of immune profiling in differentiating between these diseases. Biopsy specimens were acquired around ulcerations on the terminal ileum or cecum from five patients with each disease. Panel 1 included multiplex immunohistochemistry staining for CD8, CD4, Foxp3, CD20, programmed death-1, and granzyme B. CD56, CD68, CD163, CD11c, and HLA-DR were analyzed in panel 2. The differences in cytotoxic T cells (CD8^+^CD4^−^Fopx3^−^CD20^−^), helper T cells (CD8^−^CD4^+^Fopx3^−^CD20^−^), and regulatory T cells (CD8^−^CD4^+^Fopx3^+^CD20^−^) were also not significant. However, M1 macrophage (CD68^+^CD163^−^HLA^−^DR^−^) cell densities were significantly higher in intestinal BD than in other diseases. The expression level of dendritic cells (CD56^−^CD68^−^CD163^−^CD11c^+^HLA-DR^+^) was highest in intestinal TB and lowest in intestinal BD. The expression of immune cells, including M1 macrophages and dendritic cells, was different between CD, intestinal BD, and intestinal TB. Immune profiling can be helpful for establishing differential diagnoses of inflammatory bowel diseases.

## 1. Introduction

Inflammatory bowel disease (IBD) involves chronic, relapsing, and remitting inflammation throughout the intestine. The incidence of IBD has continued to increase globally and has recently increased in East Asian countries such as Korea, Japan, and China [1]. Inflammatory diseases of the bowel include intestinal diseases like intestinal Behcet’s disease (BD) and intestinal tuberculosis (TB), as well as Crohn’s colitis (CD) and ulcerative colitis. These diseases share many characteristics, including genetic background, clinical manifestations, and therapeutic strategies. Therefore, distinguishing between them in clinical practice is quite challenging. The misdiagnosis of intestinal TB as CD can worsen patients’ outcomes because corticosteroids or immunosuppressive drugs can aggravate intestinal TB. Conversely, misdiagnosing CD as intestinal TB can expose patients to the toxicity of anti-TB drugs and aggravate the prognosis of CD patients by delaying the diagnosis. It is important to reduce the misdiagnosis rates between chronic bowel diseases to improve patient outcomes with proper management [2]. Accordingly, new predictive models combining colonoscopic findings with laboratory and radiologic features have been developed for more precise differential diagnoses between CD and intestinal TB [3]. In addition, there have been several attempts to incorporate RNA sequencing in the diagnosis of IBD [4].

One of the pathogeneses of IBD is thought to be a defective immune reaction. The dysregulation of immune homeostasis or immune intolerance induces overactive immunoreactions, leading to chronic gastrointestinal tract inflammatory disorders [5]. Studies have proven that T lymphocytes and macrophages could be used as biomarkers for the differential diagnosis of IBD [6]. The expression of T regulatory (Treg) cells in peripheral blood and colonic mucosa was significantly higher in intestinal TB than in CD [7]. The expression of M1 macrophage cells was more predominant in CD biopsies than in intestinal TB biopsies, showing the highest expression in the colonic mucosa of granuloma-positive CD [8]. Immune profiling can also become a tool for differentiating between bowel diseases.

We previously published a study on immune profiling for colitis-associated cancer using a quantitative multispectral imaging system. This system enables the detection of multiple markers simultaneously and performs automated quantification analysis with excellent resolution [9]. Multiplex immunohistochemical (IHC) and immunofluorescence technologies are promising devices to overcome the limitations of conventional IHC by allowing multiplex analysis of numerous immune markers. This study aimed to compare the expression levels of immune markers between CD, intestinal BD, and intestinal TB and evaluate the usefulness of immune profiling in differential diagnoses using this advanced imaging system.

## 2. Methods

### 2.1. Study Population

We collected tissue samples from five patients with CD, intestinal BD, and intestinal TB enrolled at Asan Medical Center, a tertiary university hospital in Seoul, Republic of Korea. These bowel diseases were confirmatively diagnosed between 2012 and 2019 using conventional clinical, radiologic, endoscopic, and histopathologic criteria. Clinical information, including age, gender, duration from symptom onset to diagnosis, disease location and activity, and serologic C-reactive protein (CRP) and calprotectin test results at disease diagnosis, was retrieved from patients’ medical records. The study protocol was approved by the Institutional Review Board of Asan Medical Center, Seoul, Republic of Korea (no. 2019-0433), and all methods were performed in accordance with relevant guidelines and regulations.

### 2.2. Multiplex Immunofluorescence Staining

All tissues used in this study were acquired from endoscopic biopsies at the terminal ileum or ileocecal valve at the time of the diagnosis. Written informed consent was obtained from all included patients. Four tissue cultures of patients with intestinal TB were positive for Mycobacterium tuberculosis, and the remaining tissue was positive for M. bacterium according to polymerase chain reaction (PCR) analysis. Tissue samples were cut into 4 μm sections for formalin-fixed paraffin-embedded (FFPE) blocks. The slides were baked for more than 1 h in a dry oven at 60 °C, followed by multiplex immunofluorescence staining using a Leica Bond Rx™ Automated Stainer (Leica Biosystems, Wetzlar, Germany). Slides were tested against two panels of markers. Panel 1 consisted of CD8, CD4, granzyme B, programmed cell death receptor 1 (PD-1), CD 20, Foxp3, and DAPI, and panel 2 consisted of CD68, CD11c, HLA-DR, CD56, CD163, CK, and DAPI. The slides were deparaffinized using Leica Bond Dewax solution (#AR9222, Leica Biosystems) and we performed antigen retrieval using Bond Epitope Retrieval solution 2 (#AR9640, Leica Biosystems) for 30 min. Each round was sequentially reacted for 5 min with 1× antibody diluent/block (ARD1001EA, Akoya Biosciences, Marlborough, MA, USA), 30 min with primary antibody, 10 min with 1× Opal Polymer HRP Ms + Rb (ARH1001EA, Akoya Biosciences), and 10 min with Tyramide Signal Amplification (Akoya Biosciences). Then, the slides were heated at 95 °C for 20 min using Bond Epitope Retrieval solution 1 (#AR9961, Leica Biosystems) to remove antibodies and TSA from the previous round. After all steps of blocking for antigen retrieval were completed, nuclei were stained with DAPI (62248, Thermo Scientific, Waltham, MA, USA) for counterstaining. Lastly, mounting was carried out using ProLong^TM^ Gold Antifade Mountant (P36934, Thermo Scientific). The primary antibodies and corresponding TSA used for each protein were as follows: anti-CD8 (1:300, MCA1817, Bio-Rad, Hercules, CA, USA) detected by Opal 570, anti-CD4 (1:200, Ab133616, Abcam, Cambridge, UK) detected by Opal 520, anti-granzyme B (1:100, 262A, Cell marque) detected by Opal690, anti-PD-1 (1:500, Ab137132, Abcam) detected by Opal 620, CD20 (1:100, Ab9475, Abcam) detected by Opal 480, and anti-Foxp3 (1:100, Ab20034, Abcam) detected by Opal TSA-DIG and Opal 780 in panel 1; anti-CD68 (1:300, Ab192847, Abcam) detected by Opal 570, anti-CD11c (1:500, Ab52632, Abcam) detected by Opal 520, anti-HLA-DR (1:2000, Ab7856, Abcam) detected by Opal 690, anti-CD56 (1:500, Ab75813, Abcam) detected by Opal 620, anti-CD163 (1:500, Ab182422, Abcam) detected by Opal 480 in panel 2. Figure 1 and Figure 2 show representative images of the multispectral IHC staining in panels 1 and 2. The entire multiplex IHC process, including staining, scanning, and analysis, was performed in prismCDX (prismCDX Co., Ltd., Hwaseong-si, Republic of Korea).

### 2.3. Multispectral Scanning and Analysis

Multiplex stained slides were scanned at 20× magnification using a PhenoImager HT (Akoya Biosciences). Images were selected using the Phenochart^TM^ Whole Slide Viewer (Akoya Biosciences) and analyzed using inForm^®^ Tissue Analysis Software (version 2.6, Akoya Biosciences). Based on DAPI staining, each single cell was segmented, and phenotyping was performed according to the expression compartment and intensity of each marker. After designating the region (ROI, region of interest) to be analyzed on the tissue slide, the same algorithm created in this way was applied and we performed batch-running. The exported data were consolidated and analyzed in R software (version 4.2.0) using the phenoptr (Akoya Biosciences) and phenoptrReport (Akoya Biosciences) packages. We defined cytotoxic T (Tc) cells as CD8^+^CD4^−^Fopx3^−^CD20^−^, helper T (Th) cells as CD8^−^CD4^+^Fopx3^−^CD20^−^, regulatory T (Treg) cells as CD8^−^CD4^+^Fopx3^+^CD20^−^, CD8^+^ regulatory T (CD8^+^Treg) cells as CD8^+^CD4^+^Fopx3^+^CD20^−^, and B cells as CD8^−^CD4^−^Fopx3^−^CD20^+^ in panel 1. Accordingly, we defined NK cells as CD56^+^CD163^−^CD11c^−^, M1 macrophages as CD56^−^CD68^+^CD163^−^HLA-DR^−^, M2 macrophages as CD56^−^CD163^+^CD11c^−^HLA-DR^−^, dendritic cells as CD56^−^CD68^−^CD163^−^CD11c^+^HLA-DR^+^, professional antigen-presenting cells (APCs) as CD56^−^CD68^−^CD163^−^CD11c^−^HLA-DR^+^, and monocytes as CD56^−^CD68^−^CD163^−^CD11c^+^HLA-DR^−^ in panel 2.

### 2.4. Statistical Analysis

Immune cell density was compared between CD, intestinal BD, and intestinal TB and presented as cell count per mm^2^. The comparison of continuous variables between three diseases was evaluated using the Kruskal–Wallis test. *p*-values of <0.05 were considered to be statistically significant. R version 4.2.0 (R Foundation for Statistical Computing, Vienna, Austria) was used for statistical analyses.

## 3. Results

### 3.1. Baseline Characteristics of the Study Population

Five patients with CD, intestinal BD, and intestinal TB were enrolled in this study. Their baseline characteristics and clinical manifestations are shown in Table 1. The median age of the patients with intestinal TB was higher than that of the other two groups (*p* = 0.04). Only one patient with intestinal TB was taking the hypertension medication of calcium channel blockers. The enrolled patients had no other comorbidities, including histories of human immunodeficiency virus and immunosuppressive agents. The median duration from symptom onset to diagnosis in the CD and intestinal BD groups was five and three months, respectively. However, the patients in the intestinal TB group were mostly diagnosed without symptoms. The initial CRP levels at IBD diagnosis were not different between the three groups. The calprotectin levels were not different between patients with CD and intestinal BD (388 µg/g and 412 µg/g, respectively). The median CDAI score of CD patients was 52 (range 34–66), indicating mild disease activity.

### 3.2. Comparison of Immune Cell Densities in Panel 1

We evaluated the quantification of immune cells via phenotyping analysis of the multispectral imaging system. The densities of CD4^+^, CD8^+^, Foxp3^+^, CD20^+^, PD-1^+^, and granzyme B^+^ cells were not significantly different between CD, intestinal BD, and intestinal CD (Table 2). In the analysis of cell types, the quantification of Tc, Th, Treg, CD8^+^Treg, and B cells was not different between diseases (Figure 3).

### 3.3. Comparison of Immune Cell Densities in Panel 2

Table 3 shows the quantification of immune cells in panel 2 using the phenotyping method. CD68^+^ cell density was the highest in intestinal BD and lowest in CD, with statistical significance (*p* < 0.001). Other immune cells, including CD56^+^, CD163^+^, CD11c^+^, and HLA-DR^+^, were not different between the three diseases. M1 macrophage cell densities were significantly higher in intestinal BD than in CD or intestinal TB (*p* < 0.001) (Figure 4). The number of dendritic cells was highest in intestinal TB and lowest in intestinal BD (*p* = 0.012). The NK cell, M2 macrophage, professional APC cell, and monocyte densities were not different between CD, intestinal BD, and intestinal TB.

## 4. Discussion

Our study describes the immune profiling of inflammatory diseases of the bowel to identify the usefulness of differential diagnosis using a quantitative multispectral imaging system. Although a small number of tissues were included in the present study, the differences in immune cell densities, including M1 macrophages and dendritic cells, could be meaningful features.

CD, intestinal BD, and intestinal TB usually present with ulcerative lesions on the terminal ileum or cecum and ascending colon. Therefore, ileocolonoscopy is essential in establishing a diagnosis and differentiating between these diseases. The findings of a longitudinal ulcer, aphthous ulcer, the presence of rectosigmoid involvement, a cobblestone appearance, and luminal stricture favor CD, and a transverse ulcer and patulous ileocecal valve favor intestinal TB [10]. A few large, deep, round, or oval ulcers with discrete borders on the ileocecal area favor intestinal BD [11]. However, these bowel diseases share clinical manifestations, and characteristic endoscopic ulcerations are often not seen in many cases. Distinguishing these diseases with only endoscopic features is quite difficult in clinical practice. Endoscopy also has the advantage of enabling the acquisition of biopsy specimens for histologic evaluation. Architectural abnormalities, including crypt distortion, branching, and shortening, and inflammatory features, like basal plasmacytosis, transmural inflammation, focal cryptitis, and granuloma, can be observed on mucosal biopsies [12]. However, histologic findings cannot be used as an accurate diagnostic tool because of the pathologic features shared among these diseases. In suspected cases of intestinal TB, demonstrating the *Mycobacterium* organism in biopsy specimens is very difficult, and acid-fast bacillus (AFB) staining has poor sensitivity of 2.7–37.5% for diagnosing TB [13]. Also, the sensitivity of AFB culture varies from 19–70%, and the PCR analysis of AFB is not a standardized diagnostic test [14].

Along with endoscopic and histologic features, immune profiling can be another tool for making an accurate diagnosis and decreasing the misdiagnosis rate of bowel diseases. Highly differentiated T-cell subsets play a key role in the regulation and effector phase of the immune response. Tregs, critically involved in maintaining intestinal homeostasis, were decreased in inflamed mucosa like that of CD [15]. However, Tregs could be increased to suppress anti-microbial immune responses, especially against pathogens causing persistent infection. In a preliminary study, Foxp3 mRNA expression was significantly elevated in colonic biopsies obtained from intestinal TB patients compared to CD patients [16]. Another study also showed that the frequency of CD4^+^CD25^+^FOXP3^+^ Tregs in peripheral blood was significantly higher in intestinal TB than in CD [7]. These results showed excellent diagnostic accuracy, with a sensitivity of 75% and a specificity of 90%, in differentiating between CD and intestinal TB. We assumed that since T lymphocytes, including Tc, Th, and Treg cells, were important to the immune mechanisms of IBD, they could be used as immune markers to differentiate between chronic bowel diseases. However, there were no significant differences in T lymphocytes in panel 1 between the three diseases. The median Treg densities were not different between CD and intestinal TB (198 vs. 166, *p* = 0.715). T cells are known to be involved in the pathogenesis of IBD, and Tregs are detected abundantly at inflammation sites showing pro-inflammatory features. However, the role of T cells in IBD has been controversial in many studies [17]. Thus, further studies evaluating many specimens are needed to confirm the difference in T cells between inflammatory bowel diseases.

Our findings showed differences in the density of other immune cells besides T cells. CD68^+^ M1 macrophages were highly expressed in intestinal BD compared to other diseases, and dendritic cells were the highest in intestinal TB. A study suggested that M1 macrophage-predominant inflammation was associated with the pathogenesis of BD, showing that C-C chemokine receptor 1 surface expression on M1 macrophages was significantly increased in BD compared to healthy controls [18]. In another study, serum from BD patients promoted macrophage polarization toward proinflammatory M1 macrophages through nuclear factor (NF)-κB signaling [19]. Thus, M1 macrophages might be a potential therapeutic target for BD. Macrophages had plasticity between pro-inflammatory M1 and anti-inflammatory M2 macrophages. The exposure of M2 macrophages to M1 signals induced the repolarization of differentiated macrophages, which could be pursued for therapeutic goals [20]. *Mycobacterium tuberculosis* interferes with the function of dendritic cells, which are potent professional APCs. However, the outcomes of the interaction between mycobacteria and dendritic cells are still contradictory. At the onset of the inflammatory response against *M. tuberculosis*, dendritic cells are highly represented at sites of infection. As the infection progresses, M. tuberculosis inhibits dendritic cell maturation and impairs their ability to stimulate antigen-specific T cells [21]. In the present study, we analyzed immune cells in tissue specimens with active ulcerations. Our results showing the high expression of dendritic cells in intestinal TB compared to other diseases could be considered to reflect an early-stage inflammatory response.

Immune profiling has been especially used in cancer to predict prognoses and provide targets for immunotherapy [22]. The ability to assess the relationships of immune cells with a patient’s cancer is useful for predicting their prognosis, including disease-free survival (DFS) and overall survival (OS), and represents a critical tool in evaluating therapeutic options for that patient. Recently, studies on immune markers have expanded their scope to other fields. The immune profile in viral disease helps to detect chemokines and cytokines, which play a crucial role in immunopathology. It is important to identify the immune responses that offer protection against infection, which can be utilized to guide the development of vaccines. Consequently, it serves as the basis for optimum disease management and drug or vaccine design [23]. The profile of immune cells has been actively studied to understand the pathogenesis of IBD because this is postulated to result from immune dysregulation in response to environmental triggers in genetically susceptible individuals [24]. Improved analysis of immune cell landscapes in intestinal tissues may determine new therapeutic methods that can be tailored to the disease type. Our previous study evaluated the characteristics of immune profiling in IBD-associated cancer. It showed that colitis-associated cancer had different levels of immune marker expression compared with those of sporadic colorectal cancer, suggesting possible treatment targets for diseases associated with IBD [9]. A greater understanding of the gut mucosal immune cell metabolism leads to the elucidation of the pathogenesis of IBD and the identification of disease characteristics. Our findings in the current study could be helpful in differentiating between disease phenotypes with specific immune cell features. Also, integrating immune profiling with clinical, endoscopic, and serologic parameters could contribute to developing a qualified prediction model. Any combination or equation of two or more pieces of information is helpful for establishing accurate diagnoses in clinical settings to identify patients at great risk.

Single-cell technologies could lead to a better understanding of IBD complexity by integrating single-cell RNA sequencing and spatial molecular imaging analysis from colonic tissue samples, which showed the heterogeneity of macrophages and neutrophils in the inflamed colonic mucosa of IBD patients [25]. Our study used multiplex immunofluorescence assays with tyramine signal amplification staining, which can simultaneously detect multiple immune markers in the same tissue section. This novel device performs multiplex IHC staining, multispectral image acquisition, and analysis. In addition, it provides quantitative data on the (co-)expression levels and spatial localization of immune cell subtypes using digital image analysis software, offering high-quality throughput. With these advantages, this revolutionary technology has been used to discover tumor microenvironments, new targets for treatment, and prognostic and predictive biomarkers, and to conduct translational studies [26]. Many cancer studies have used multiplex immunofluorescence assays for highly reproducible, efficient, and cost-effective tissue studies. A recent study demonstrated that this method appeared to be associated with improved performance in predicting responses to programmed cell death ligand 1/PD-1 treatment in different solid tumor types [27]. Our study was conducted in non-malignant tissues to characterize the different types of immune cell populations. The ability to label multiple markers on a single section is of particular significance when study samples are taken from rare cases with low availability status. Chronic colitis can include a variety of inflammatory conditions or tissue depletion. The emerging multiplex IHC technologies are promising in inflammatory diseases and provide comprehensive information about cell composition and spatial arrangement in clinical settings.

This study had a few limitations. A small number of patients were included, and a healthy control was not included. This may have resulted in selection bias. Next, this study was conducted in a retrospective and non-randomized manner at a single center. Biases due to unrecognized or unmeasured factors might have occurred. Third, we examined tissue samples acquired at the time of diagnosis at our hospital, which was a tertiary referral hospital. Previous treatments, including the use of steroids or 5-aminosalicylate, could be correlated with immune cell expression. Fourth, gut microbiota could contribute to the substantial changes in immune cell composition in IBD. Our study did not consider the influence of microbiota. Lastly, most of the tissue samples were acquired on the terminal ileum. However, one colonic tissue taken from the ileocecal valve of an intestinal BD patient was included. In addition, patients with intestinal TB were older than patients with other diseases. These could be confounding factors in our results.

## 5. Conclusions

The immune profiling of inflammatory bowel diseases was helpful in differentiating between disease subtypes. M1 macrophages and dendritic cells can be immune markers for making precise diagnoses of CD, intestinal BD, and intestinal TB. Further studies analyzing many specimens are needed to support the results of this study.

## Figures and Tables

**Figure 1 diagnostics-13-02904-f001:**
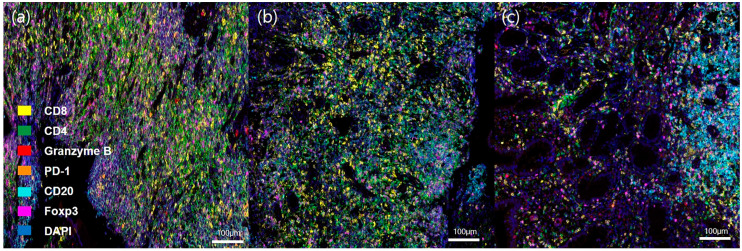
Representative images of multispectral immunohistochemical staining in panel 1 (20×). (**a**) CD. (**b**) Intestinal BD. (**c**) Intestinal TB.

**Figure 2 diagnostics-13-02904-f002:**
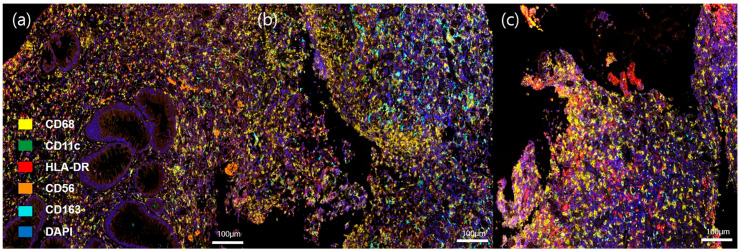
Representative images of multispectral immunohistochemical staining in panel 2 (20×). (**a**) CD. (**b**) Intestinal BD. (**c**) Intestinal TB.

**Figure 3 diagnostics-13-02904-f003:**
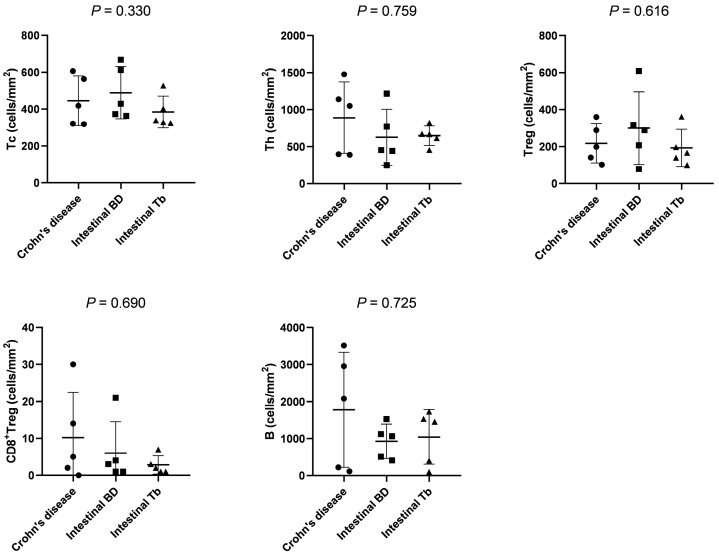
Comparison of immune cell densities between CD, intestinal BD, and intestinal TB in panel 1. Abbreviations: Tc, cytotoxic T cell; Th, helper T cell; Treg, regulatory T cell.

**Figure 4 diagnostics-13-02904-f004:**
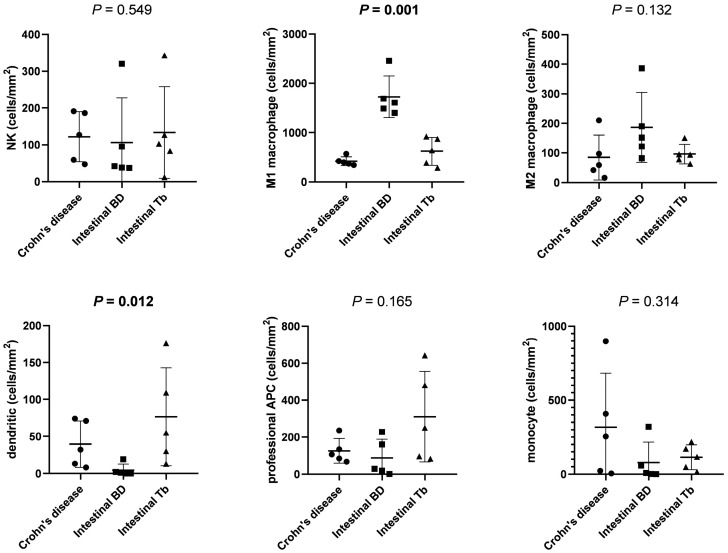
Comparison of immune cell densities between CD, intestinal BD, and intestinal TB in panel 2. The densities of M1 macrophage and dendritic cells were significantly different between the three groups. Abbreviations: NK, natural killer cell; APC, antigen-presenting cell.

**Table 1 diagnostics-13-02904-t001:** Baseline characteristics of patients with Crohn’s disease, intestinal Behcet’s disease, and intestinal tuberculosis.

Variables	Crohn’s Disease (*n* = 5)	Intestinal BD (*n* = 5)	Intestinal TB (*n* = 5)
Male, *n* (%)	3 (60)	3 (60)	2 (40)
Age at diagnosis, median year (range)	32 (16–40)	34 (30–57)	49 (35–72)
Duration from symptom onset to diagnosis, median month (range)	5 (0–5)	3 (2–10)	0 (0–3)
Initial median CRP, mg/dL (range)	0.6 (0.2–2)	0.4 (0.1–6)	0.2 (0.1–0.2)
Initial median calprotectin, µg/g (range)	388 (43.1–566)	412 (202–457)	N/A
Severity (CDAI)	52 (34–66)		

BD, Behcet’s disease; TB, tuberculosis; CRP, C-reactive protein; N/A, not available; CDAI, Crohn’s disease activity index.

**Table 2 diagnostics-13-02904-t002:** Comparison of immune cell densities in panel 1 between Crohn’s disease, intestinal Behcet’s disease, and intestinal tuberculosis.

	Crohn’s Disease	Intestinal BD	Intestinal TB	*p*-Value
CD4^+^, median cells/mm^2^ (range)	2115 (510–3085)	783 (375–2019)	1358 (587–1682)	0.620
CD8^+^, median cells/mm^2^ (range)	503 (330–1125)	503 (390–959)	481 (339–694)	0.690
Foxp3^+^, median cells/mm^2^ (range)	339 (131–552)	439 (99–755)	245 (120–526)	0.725
CD20^+^, median cells/mm^2^ (range)	3176 (129–4971)	1281 (437–1724)	1938 (122–2652)	0.650
PD-1^+^, median cells/mm^2^ (range)	189 (12–317)	40 (20–101)	108 (12–197)	0.392
Granzyme B^+^, median cells/mm^2^ (range)	120 (98–222)	250 (28–411)	214 (80–467)	0.365

BD, Behcet’s disease; TB, tuberculosis.

**Table 3 diagnostics-13-02904-t003:** Comparison of immune cell densities in panel 2 between Crohn’s disease, intestinal Behcet’s disease, and intestinal tuberculosis.

	Crohn’s Disease	Intestinal BD	Intestinal TB	*p*-Value
CD56^+^, median cells/mm^2^ (range)	199 (63–263)	75 (44–335)	160 (14–602)	0.582
CD68^+^, median cells/mm^2^ (range)	530 (508–840)	1828 (1648–3366)	1224 (508–1404)	<0.001
CD163^+^, median cells/mm^2^ (range)	63 (17–228)	154 (89–408)	114 (69–167)	0.165
CD11c^+^, median cells/mm^2^ (range)	362 (20–1160)	19 (0–616)	304 (36–649)	0.252
HLA-DR^+^, median cells/mm^2^ (range)	258 (193–451)	152 (3–640)	890 (168–1147)	0.208

BD, Behcet’s disease; TB, tuberculosis.

## Data Availability

The data presented in this study are available on request from the corresponding author. The data are not publicly available due to privacy or ethical restrictions.

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
