# Peer review of "Clinical Usefulness of Immune Profiling for Differential Diagnosis between Crohn’s Disease, Intestinal Tuberculosis, and Behcet’s Disease"

_diagnostics, 2023, doi:10.3390/diagnostics13182904_

Round 1
Reviewer 1 Report
The authors do not describe the comorbidity profile of the patients. Was the presence of HIV ruled out? Was the presence of immunodeficiency ruled out as the common variable, IgA deficiency or others? Drugs that could potentially influence (olmesartan, steroids, PPI,...) are not included.
Author Response
|
Response to Reviewer 1 Comments
|
||
|
1. Summary |
|
|
|
We thank reviewer 1 for providing a constructive review and comments. Please find the detailed responses below and the corresponding revision highlighted in track change mode in the re-submitted file.
|
||
|
2. Questions for General Evaluation |
Reviewer’s Evaluation |
Response and Revisions |
|
Does the introduction provide sufficient background and include all relevant references? |
Yes |
|
|
Are all the cited references relevant to the research? |
Yes |
|
|
Is the research design appropriate? |
Can be improved |
|
|
Are the methods adequately described? |
Yes |
|
|
Are the results clearly presented? |
Yes |
|
|
Are the conclusions supported by the results? |
Yes |
|
|
3. Point-by-point response to Comments and Suggestions for Authors |
||
|
Comments 1: The authors do not describe the comorbidity profile of the patients. Was the presence of HIV ruled out? Was the presence of immunodeficiency ruled out as the common variable, IgA deficiency or others? Drugs that could potentially influence (olmesartan, steroids, PPI,...) are not included.
Response 1: We thank the reviewer for this comment. We checked other comorbidities of enrolled patients and confirmed that there were no specific immunodeficiency states. We have added the following contents to the RESULTS section (revised manuscript, page 4, lines 154–156) as follows: “Only one patient with intestinal TB was taking hypertension medication of calcium channel blockers. There were no other comorbidities including histories of human immunodeficiency virus and immunosuppressive agents in enrolled patients.” |
||
|
|
||
Reviewer 2 Report
Dear Authors
The well designed study and well-presented results is qualified sufficiently for publication without any revision.
All parts of the manuscript have been presented in a standard format.
Sincerely
Author Response
We thank reviewer 2 for providing comments. We hope that our research will be helpful in the diagnosis of inflammatory bowel diseases. Thank you.
Reviewer 3 Report
This is a very elegant report that aims to elucidate the differential diagnosis of inflammatory bowel disease (IBD) by combining a multiplex platform that employs both immunofluorescence and immunohistochemistry.
Major points:
* In the introduction, the authors should mention that there have been several attempts trying to incorporate RNA-Seq in the diagnosis of IBD. For example, Massimino L, Lamparelli LA, Houshyar Y, D’Alessio S, Peyrin-Biroulet L, Vetrano S, et al.. The Inflammatory Bowel Disease Transcriptome and Metatranscriptome Meta-Analysis (IBD TaMMA) Framework. Nat Comput Sci (2021) 1(8):511–5. doi: 10.1038/s43588-021-00114-y.
* In the Discussion, the authors should mention how their own results could be exploited in the context of novel technologies such as CosMx spatial genomics. For example, please see: PMID: 37495570
* The authors should further discuss potential confounders: anatomical regions and age. They mentioned: "terminal ileum or ilocecal valve" (lines 84-85), Can there be differences in innate and adaptive immune signature according to the exact site? Patients with intestinal TB are older (49 vs. early thirties). Can this affect the cellular composition of the gut? In the absence of healthy controls, this can be a significant confounder.
* The authors forgot to mention the contribution of the microbiota to the changes in cell type composition. Maybe they should include this as a limitation or another confounder in the discussion.
* The reader will benefit if, in the Discussion, the authors mention briefly the plasticity between M1/M2 polarization and some examples.
Minor points:
* The CMV abbreviation on line 164 shows up only once.
* Figures 1 and 2 need a scale (in addition to the 20x mention).
* It could be more informative to show the individual patients' points in Figures 3 and 4 (overlaid on the mean and IQR).
* The authors need to include corrections to multiple comparisons and adjust the p-values accordingly (throughout the entire manuscript). For example, the reviewer wonders whether the p-values shown in red in Figure 4 still hold true after accounting for multiple comparisons.
Author Response
|
Response to Reviewer 3 Comments
|
||
|
1. Summary |
|
|
|
We thank reviewer 3 for providing a constructive review and comments. Please find the detailed responses below and the corresponding revision highlighted in track change mode in the re-submitted file.
|
||
|
2. Questions for General Evaluation |
Reviewer’s Evaluation |
Response and Revisions |
|
Does the introduction provide sufficient background and include all relevant references? |
Can be improved |
|
|
Are all the cited references relevant to the research? |
Can be improved |
|
|
Is the research design appropriate? |
Yes |
|
|
Are the methods adequately described? |
Yes |
|
|
Are the results clearly presented? |
Can be improved |
|
|
Are the conclusions supported by the results? |
Yes |
|
|
3. Point-by-point response to Comments and Suggestions for Authors |
||
|
Comments 1 (Major): In the introduction, the authors should mention that there have been several attempts trying to incorporate RNA-Seq in the diagnosis of IBD. For example, Massimino L, Lamparelli LA, Houshyar Y, D’Alessio S, Peyrin-Biroulet L, Vetrano S, et al. The Inflammatory Bowel Disease Transcriptome and Metatranscriptome Meta-Analysis (IBD TaMMA) Framework. Nat Comput Sci (2021) 1(8):511–5. doi: 10.1038/s43588-021-00114-y.
Response 1: We thank the reviewer for these comments. As you recommended, we added the contents to the Instruction section (revised manuscript, page 2, lines 53–55) as follows: “In addition, there have been several attempts to incorporate RNA-sequencing in the diagnosis of IBD [4].” |
||
|
Comments 2 (Major): In the Discussion, the authors should mention how their own results could be exploited in the context of novel technologies such as CosMx spatial genomics. For example, please see: PMID: 37495570.
Response 2: We thank the reviewer for this comment. Our study performed the profiling of immune cells in the inflamed colonic mucosa with novel technologies of multiplexed immunofluorescence assays. We have added the contents of the example in the DISCUSSION section (revised manuscript, page 8, lines 286–289) as follows: “Single-cell technologies could allow to bring more understanding of IBD complexity by integrating single-cell RNA sequencing and spatial molecular imaging analysis from colonic tissue samples [25]. The study showed the heterogeneity of macrophages and neutrophils in the inflamed colonic mucosa of IBD patients.”
Comments 3 (Major): The authors should further discuss potential confounders: anatomical regions and age. They mentioned: "terminal ileum or ilocecal valve" (lines 84-85), Can there be differences in innate and adaptive immune signature according to the exact site? Patients with intestinal TB are older (49 vs. early thirties). Can this affect the cellular composition of the gut? In the absence of healthy controls, this can be a significant confounder.
Response 3: Thank you very much for this comment. Our study had several confounding factors. We have added this limitation in the DISCUSSION section (revised manuscript, page 9, lines 316–319) as follows: “Lastly, most of the tissue samples were acquired on the terminal ileum. However, one colonic tissue taken from the ileocecal valve of an intestinal BD patient was included. In addition, patients with intestinal TB were older than patients with other diseases. These could be confounding factors in our results.”
Comments 4 (Major): The authors forgot to mention the contribution of the microbiota to the changes in cell type composition. Maybe they should include this as a limitation or another confounder in the discussion.
Response 4: We thank the reviewer for this comment. Our study had the limitation of not considering microbiota. We have added this limitation in the DISCUSSION section (revised manuscript, page 9, lines 314–316) as follows: “Fourth, gut microbiota could contribute the substantial changes of immune cell composition in IBD. Our study did not consider the influence of microbiota.”
Comments 5 (Major): The reader will benefit if, in the Discussion, the authors mention briefly the plasticity between M1/M2 polarization and some examples.
Response 5: We thank the reviewer for this comment. We added the content of the plasticity of M1/M2 macrophages in the DISCUSSION section (revised manuscript, page 7, lines 250–253) as follows: “Macrophages had plasticity between pro-inflammatory M1 and anti-inflammatory M2 macrophages. Exposure of M2 macrophage to M1 signals induced repolarization of differentiated macrophages, which could be pursued for therapeutic goals [20].”
Comments 1 (Minor): The CMV abbreviation on line 164 shows up only once.
Response 1: Thank you very much for knowing our mistakes. We deleted the CMV abbreviation in the table 1.
Comments 2 (Minor): Figures 1 and 2 need a scale (in addition to the 20x mention).
Response 2: Thank you very much for this comment. We added scales in the figures 1 and 2. Figure 1. Representative images of multispectral immunohistochemical staining in panel 1. (20X) (a) CD. (b) intestinal BD. (c) intestinal TB. Figure 2. Representative images of multispectral immunohistochemical staining in panel 2. (20X) (a) CD. (b) intestinal BD. (c) intestinal TB.
Comments 3 (Minor): It could be more informative to show the individual patients' points in Figures 3 and 4 (overlaid on the mean and IQR).
Response 3: We thank the reviewer for this comment. We changed Figures 3 and 4 to the individual patients’ points. Figure 3. Comparison of immune cell densities between CD, intestinal BD, and intestinal TB in panel 1. Figure 4. Comparison of immune cell densities between CD, intestinal BD, and intestinal TB in panel 2.
Comments 4 (Minor): The authors need to include corrections to multiple comparisons and adjust the p-values accordingly (throughout the entire manuscript). For example, the reviewer wonders whether the p-values shown in red in Figure 4 still hold true after accounting for multiple comparisons.
Response 4: We thank the reviewer for this comment. We evaluated each immune cell by the Kruskall-Wallis test. We could not evaluate those cells by multiple comparisons because small number of patients included. This is our major limitation and wrote the content as the first limitation. |
||